# An Association between OXPHOS-Related Gene Expression and Malignant Hyperthermia Susceptibility in Human Skeletal Muscle Biopsies

**DOI:** 10.3390/ijms25063489

**Published:** 2024-03-20

**Authors:** Leon Chang, Rebecca Motley, Catherine L. Daly, Christine P. Diggle, Philip M. Hopkins, Marie-Anne Shaw

**Affiliations:** 1Leeds Institute of Medical Research at St James’s, University of Leeds, Leeds LS9 7TF, UK; l.chang@leeds.ac.uk (L.C.); c.p.diggle@leeds.ac.uk (C.P.D.); p.m.hopkins@leeds.ac.uk (P.M.H.); 2Malignant Hyperthermia Unit, St James’s University Hospital, Leeds LS9 7TF, UK

**Keywords:** malignant hyperthermia, skeletal muscle, mitochondria, oxidative phosphorylation, RNA sequencing, gene expression

## Abstract

Malignant hyperthermia (MH) is a pharmacogenetic condition of skeletal muscle that manifests in hypermetabolic responses upon exposure to volatile anaesthetics. This condition is caused primarily by pathogenic variants in the calcium-release channel RYR1, which disrupts calcium signalling in skeletal muscle. However, our understanding of MH genetics is incomplete, with no variant identified in a significant number of cases and considerable phenotype diversity. In this study, we applied a transcriptomic approach to investigate the genome-wide gene expression in MH-susceptible cases using muscle biopsies taken for diagnostic testing. Baseline comparisons between muscle from MH-susceptible individuals (MHS, *n* = 8) and non-susceptible controls (MHN, *n* = 4) identified 822 differentially expressed genes (203 upregulated and 619 downregulated) with significant enrichment in genes associated with oxidative phosphorylation (OXPHOS) and fatty acid metabolism. Investigations of 10 OXPHOS target genes in a larger cohort (MHN: *n* = 36; MHS: *n* = 36) validated the reduced expression of *ATP5MD* and *COQ6* in MHS samples, but the remaining 8 selected were not statistically significant. Further analysis also identified evidence of a sex-linked effect in *SDHB* and *UQCC3* expression, and a difference in *ATP5MD* expression across individuals with MH sub-phenotypes (trigger from in vitro halothane exposure only, MHS_h_ (*n* = 4); trigger to both in vitro halothane and caffeine exposure, MHS_hc_ (*n* = 4)). Our data support a link between MH-susceptibility and dysregulated gene expression associated with mitochondrial bioenergetics, which we speculate plays a role in the phenotypic variability observed within MH.

## 1. Introduction

Variants in the calcium release channel, ryanodine receptor isoform 1 (RYR1), are associated with malignant hyperthermia (MH), a pharmacogenetic disorder of skeletal muscle. Individuals who are susceptible to malignant hyperthermia (MH) suffer severe hypermetabolic reactions when exposed to general anaesthesia, which can be fatal. This highlights the importance of presurgical screening and testing for familial pathogenic variants. The European Malignant Hyperthermia Group (EMHG) currently recognises an ever-growing list of 66 pathogenic/likely pathogenic variants in *RYR1* and two in *CACNA1S* (encodes α-subunit of the dihydropyridine receptor) for MH diagnosis [1]. Mutations within these genes result in calcium dysregulation and are present in approximately 76% of UK MH patients, whilst other patients do not have a known major causative variant [2]. Our current knowledge of MH genetics is incomplete.

The UK adopts the EMHG guidelines, achieving MH diagnosis through a combination of genetic blood testing of family members, where possible, and, for the majority of cases, laboratory investigation using the in vitro contracture test (IVCT) [3]. Potential MH-susceptible candidates, with no known familial variant, are referred for IVCT, where skeletal muscle biopsies are stretched to optimum length with electrical stimulation and treated with incremental doses of halothane and caffeine in vitro. This treatment triggers a contracture response in MH-susceptible muscle fibres, and the strength of these contractures is measured for diagnostic purposes. IVCT MH-susceptible (MHS) phenotypes are variable between patients with some responding only to halothane exposure (MHS_h_), to both halothane and caffeine (MHS_hc_), or on rare occasions only to caffeine (MHS_c_). Diagnosis as ‘affected’ or ‘unaffected’ is conservative both using genetics, where there is considerable within-family genotype-phenotype discordance, and also using the IVCT. The underlying pathology behind the observed phenotypic variation remains to be elucidated.

Studies have shown that muscle biopsies containing different *RYR1* variants exhibit varying sensitivity and response times to the IVCT, introducing the concept of weak and strong variants [4,5]. The biological consequences of specific *RYR1* variants have been investigated using different knock-in mouse models [6,7,8,9]. Mitochondrial dysfunction has been a consistent feature in MH-associated RYR1-knock in mouse models and has been speculated to be a potential influencer of the MH muscle phenotype [7,10,11,12,13,14]. Although limited, this parallels data from studies of human MHS muscle that have shown evidence of mitochondrial structure abnormality using electron microscopy [15], and functional deficits detected through high-resolution respirometry [16].

We have previously published transcriptomics data obtained from the MH-associated G2435R-RYR1 mouse model, presenting evidence of downregulated gene expression in fatty acid oxidation pathways in the homozygous genotype [14]. Furthermore, our recent publication on exertional heat illness (a closely related condition), included whole blood transcriptomics from MHS patients and showed downregulated gene expression in oxidative phosphorylation (OXPHOS) pathways [17]. These findings show that *RYR1* variants can influence gene expression in metabolic pathways associated with mitochondria, observations not yet explored using human MHS muscle samples. Here, we present genome-wide transcriptome data comparing human MH negative/normal (MHN) and MHS muscle biopsies to help identify candidate genes and pathways for further research into underlying MH pathology and genetics.

## 2. Results

### 2.1. Transcriptome Analysis

We initially investigated differential gene expression with both MHS sub-phenotypes combined (MHS_h_ and MHS_hc_), and this cohort will hereby be referred to as the ‘MHS’ phenotype. Gene expression analysis comparing MHN (*n* = 4) and MHS (*n* = 8) muscle transcriptomes revealed 822 differentially expressed genes at baseline. From this gene list, 203 (24.7%) were upregulated in MHS samples with an effect size between 0.22 and 5.17 log2 fold change (log2FC). The remaining 619 genes (75.3%) had reduced expression in MHS samples with an effect size between –0.19 and –2.06 log2FC. Functional annotation of downregulated genes showed enrichment in pathways associated with OXPHOS, adipogenesis, and fatty acid metabolism (Figure 1).

Gene ontology findings were consistent with the pathway analysis, highlighting enriched ontology terms surrounding mitochondrial function (Table 1). Notably, several ontology terms corresponded to the activity of NADH dehydrogenase—complex I of the electron transport chain (ETC) and 18 complex I-encoded genes were found downregulated in MHS muscle (between −0.82 and −0.52 log2FC, adjusted *p*-value < 0.1, Appendix A). In contrast, pathway analysis of the upregulated genes showed functional enrichment in a single inflammatory pathway—“TNF-alpha Signalling via NF-kB” in the MHS cohort. Gene ontology results for this gene set contained terms associated with RNA processing, binding, and splicing activity (Appendix A).

### 2.2. MHS Sub-Phenotypes

Next, we explored differences between the MHS sub-phenotypes, comparing MHN samples (*n* = 4) to MHS_h_ (*n* = 4) and MHS_hc_ (*n* = 4) samples, separately. This analysis showed 1654 and 68 differentially expressed genes, respectively. A large proportion of the genes were downregulated in the MHS_h_ phenotype (1281/1654, 74.5%), with similar enrichment analysis results to the MHN vs. MHS comparison. Genes downregulated in the MHS_h_ group were significantly enriched in pathway and ontology terms involved with OXPHOS, adipogenesis and mitochondrial function, whilst upregulated genes indicated RNA processing, binding, and splicing functionality (Appendix A).

We observed fewer enrichment terms in the MHN vs. MHS_hc_ comparison—most likely due to the reduced power with only 68 genes for analyses from fewer samples. The most significantly enriched upregulated pathway in the MHS_hc_ phenotype was the ‘TNF-alpha Signalling via NF-kB’ pathway, whilst downregulated pathways featured ‘Oxidative phosphorylation’ as the top hit—though this was not statistically significant (adjusted *p*-value 0.091) (Appendix A). No ontology terms were significantly enriched for genes up or downregulated in MHS_hc_ samples (Appendix A).

To identify genes shared between both MHS sub-phenotypes, we cross-compared the results from the ‘MHS_h_ vs. MHN’ and ‘MHS_hc_ vs. MHN’ gene lists. This analysis revealed 18 genes differentially expressed in both sub-phenotypes with respect to MHN controls (Figure 2). The most highly expressed genes in this list include *PLA2G2A*, *ADIPOQ*, *MAOA*, and *SCD*, which are functionally involved in fatty acid metabolism and mitochondrial function.

### 2.3. Validation of OXPHOS-Associated Genes

The enrichment of OXPHOS-related genes warranted further investigation, and we chose 10 genes to validate our findings using real-time quantitative polymerase chain reaction assays (RT-qPCR). The 10 genes selected were differentially expressed in the “MHN vs. MHS” baseline comparison, and we investigated their gene expression levels in a larger sample cohort (MHN *n* = 36. MHS *n* = 36). Within the ETC components of interest, five genes were selected based on the lowest *p*-values calculated (*ATP5MD*, *MALSU1*, *COQ6*, *NDUFC2*, *MPC2*) and another five were selected based on the largest log2FC observed (*COX8A*, *UQCC3*, *SDHB*, *NDUFA11*, *ATP5F1D*). These genes covered each complex of the ETC: complex I (*NDUFC2*, *NDUFA11*), complex II (*SDHB*), complex III (*UQCC3*), complex IV (*COX8A*), and complex V (*ATP5MD*, *ATP5F1D*) and several intermediary components (*COQ6*, *MPC2*, *MALSU1*) to give an overview of the system. Please refer to Appendix A for the assays used.

Compared to MHN, our results showed significantly reduced *ATP5MD* (*p* = 0.021) and *COQ6* (*p* = 0.049) gene expression in MHS samples (Figure 3A). The remaining eight genes showed a minor reduction in expression levels, but they were not statistically significant in the larger sample cohort: *MALSU1* (*p* = 0.593), *NDUFC2* (*p* = 0.135), *MPC2* (*p* = 0.354), *COX8A* (*p* = 0.402), *UQCC3* (*p* = 0.620), *SDHB* (*p* = 0.055), *NDUFA11* (*p* = 0.192) and *ATP5F1D* (*p* = 0.216). The log2FC effect sizes from our RT-qPCR data were much smaller than the RNA sequencing (RNAseq) counterparts, which may reflect a combination of increased biological variability within the cohort and sample IVCT phenotype (Figure 3B). The mean IVCT contracture response at 2% halothane was 0.914 g in the RNAseq-MHS samples, whilst the IVCT halothane response in RT-qPCR-MHS samples was 0.747 g.

Upon further investigation, *ATP5MD* gene expression also differed between MHS sub-phenotypes (Figure 4A), with significantly lower expression in the MHS_hc_ phenotype (Kruskal–Wallis *p* = 0.026, MHN vs. MHS_h_ *p* = 0.852, MHN vs. MHS_hc_ *p* = 0.021). This feature was not observed in the other genes investigated. Furthermore, we investigated whether sex groups influenced gene expression in these samples. Overall, we found that MHS male samples had the lowest gene expression compared to other groups, and this was statistically significant in the expression of *SDHB* (*p* = 0.022) and *UQCC3* (*p* = 0.016) (Figure 4B). Post hoc pairwise comparisons showed significantly lower *UQCC3* expression in MHS males vs. MHS females (*p* = 0.009), and lower *SDHB* expression in MHS males vs. MHN females (*p* = 0.021). No sex-associated differences were detected in the other eight genes of interest: *ATP5F1D* (*p* = 0.288), *ATP5MD* (*p* = 0.074), *COQ6* (*p* = 0.093), *COX8A* (*p* = 0.604), *NDUFA11* (*p* = 0.055), *NDUFC2* (*p* = 0.422), *MALSU1* (*p* = 0.820), *MPC2* (*p* = 0.661).

### 2.4. Effects of IVCT on Global Gene Expression

In addition to baseline differences, we also looked at the effects of IVCT on gene expression. MHN muscle biopsies (baseline *n* = 4; IVCT-halothane *n* = 4; IVCT-caffeine *n* = 4) featured 2905 and 1554 genes differentially expressed in response to IVCT-halothane and caffeine exposure, respectively. The overlapping MHN-IVCT responses (halothane and caffeine) shared elevated expression in 731 genes, which were enriched in several pathways involved in pro-inflammatory responses (Appendix A). In contrast, 467 genes were downregulated in the overlapping MHN-IVCT responses and were enriched in the “Oxidative Phosphorylation”, “Myc Targets V1”, and “Fatty Acid Metabolism” pathways.

Surprisingly, the IVCT-induced gene expression changes were less pronounced in the MHS cohort (baseline *n* = 4; IVCT-halothane *n* = 4; IVCT-caffeine *n* = 4), featuring a smaller set of differentially expressed genes (IVCT-halothane: 325 genes and IVCT-caffeine: 583 genes). The upregulated genes in the shared MHS-IVCT responses (halothane and caffeine: 235 genes) showed functional enrichment in pro-inflammatory response pathways like the MHN-IVCT response. No functional enrichment was observed in the downregulated genes shared between the MHS-IVCT responses—likely due to a lack of statistical power from the low number of gene entries (56 genes).

## 3. Discussion

In this study, we performed bulk RNA sequencing on human muscle biopsies to explore transcriptomic differences between MH phenotypes at baseline, and in response to IVCT. As our knowledge of the genetics of MH is incomplete, this approach has the potential to identify additional contributory loci in addition to increasing our knowledge surrounding pathological processes. Participants were selected considering age, sex (all male), strength of IVCT response, and presence of a pathogenic *RYR1* variant. MH susceptibility is a relatively rare condition, with limited IVCT material, precluding further standardisation, e.g., the strength of specific *RYR1* variants. Rather than specific *RYR1* variants, we opted to include human MHS samples containing a range of different *RYR1* variants, to identify gene expression profiles that are more applicable to the general MHS population.

Compared to MHN muscle, there was an overall decrease in MHS muscle gene expression at baseline. Approximately 75% of differentially expressed genes were detected at lower expression levels in the MHS cohort, with functional associations to OXPHOS, adipogenesis, and fatty acid metabolism. These pathways are highly dependent on mitochondrial function, which is speculated to be a downstream consequence of RyR1 dysfunction in MH, since MH events result in sustained muscle contractures dependent on ATP. Many of the downregulated OXPHOS genes we identified have been associated with metabolic conditions such as type 2 diabetes [18,19,20,21]. Of interest are the genes encoding NADH dehydrogenase (mitochondrial complex I) abundant in our differential gene expression results.

NADH dehydrogenase is the largest component of the ETC, composed of 44 subunits, and is responsible for the oxidation of NADH and the reduction of ubiquinone to ubiquinol through electron transfer [22]. The prevalence of mitochondrial disease is low, estimated to occur between 1 and 8000 in adults. Among the different components of the ETC, deficiencies in NADH dehydrogenase are the most frequent single enzyme deficiency responsible for mitochondrial disorders [23,24]. The numerous downregulated complex I-encoding genes identified in our sequencing data suggest an element of complex I deficiency is present within MHS skeletal muscle.

In support of these data, associations between MH muscle and complex I deficiency have previously been reported in porcine models of MH. In a study of different pig breeds, Liu and colleagues identified reduced complex I-associated gene expression and enzyme activity in the skeletal muscle of Pietrain pigs, which were homozygous-positive for MH [25]. At the transcriptional level, this group also showed lower expression of lactate dehydrogenase B and several other subunits of OXPHOS complexes, including complex I, complex II, complex IV, and ATP synthase. Furthermore, a study on anaesthetic-induced neurotoxicity reported hypersensitivity to volatile anaesthetics in *Drosophila melanogaster* carrying mutations in mitochondrial complex I, which may be relevant in the context of MH [26].

It has been hypothesized that the mitochondrial dysfunction observed in MHS muscle may be a consequence of the elevated intracellular [Ca^2+^] and increased oxidative stress associated with *RYR1* variants. Increased oxidative stress and the accumulation of reactive oxygen species (ROS) have been consistently reported in the muscle of various MH mouse models [10,13,14,27]. The interplay between ROS and Ca^2+^ has an important role in regulating cellular metabolism and its dysregulation has detrimental effects on mitochondria [28]. High intracellular [Ca^2+^] is known to cause SR stress and increased Ca^2+^ influx into mitochondria, which drives excessive ROS production. Chronic exposure to elevated ROS may explain the structural damage and functional deficiencies observed in MHS mitochondria.

A study by Thompson et al. described impaired aerobic metabolism and reduced ATP production in MHS individuals [29]. This aligned with our previously published respirometry data showing reduced OXPHOS capacity and CII deficiency in MHS mitochondria when compared to MHN, indicating functional impairment [16]. Patients susceptible to MH typically do not exhibit clinical phenotypes in the absence of anaesthetic triggers but some have reported muscle-related symptoms such as weakness, exercise intolerance, and exercise-induced rhabdomyolysis [30,31,32]. It is likely that specific MH-associated *RYR1* variants confer varying degrees of influence on mitochondrial function, and myopathic features may only be associated with more disruptive *RYR1* variants.

Although numerous OXPHOS-related genes had reduced expression levels, we noticed from our data that their effect sizes were relatively small (<1 fold change). To validate these findings, we selected 10 OXPHOS-related genes and investigated their gene expression in a larger patient cohort of a wider age range containing both male and female samples. These genes were encoded for proteins across complexes I to V of the ETC and several associated proteins that facilitate ETC function. Our validation study was able to confirm a significant reduction in *ATP5MD* and *COQ6* gene expression in the wider MHS cohort. Additionally, we identified baseline differences between MHS sub-phenotypes, with *ATP5MD* gene expression significantly downregulated in MHS_hc_ but not in MHS_h_ samples.

*ATP5MD* codes for a membrane subunit of ATP synthase (complex V) and is thought to have a role in maintaining complex V in mitochondria [33,34]. *ATP5MD* (also known as *USMG5*) was first described as a novel protein downregulated in diabetic rats [35], and histological reports have shown increased localisation in cells with high aerobic metabolism, favouring oxidative fibres in skeletal muscle [36]. Likewise, coenzyme Q6 monooxygenase (CoQ6) encoded by COQ6 is also an essential ETC component and takes part in producing coenzyme Q10 (CoQ10), relaying electrons from complexes I and II to complex III. CoQ10 is an antioxidant that acts as a cofactor for mitochondrial dehydrogenases and contributes towards the regulation of apoptosis. Severe deficiency of CoQ10 has been associated with lower ATP turnover and normal ROS production, whilst the consequence of minor deficiency is high ROS production and normal ATP levels—indicating a complex bioenergetic mechanism [37].

Results for the remaining eight loci were consistent with a reduction in expression in MHS samples but were not statistically significant, including *NDUFC2* encoding complex I. This may be due to a reduction in power associated with increased variability in the sample cohort with regard to age and sex. We subsequently conducted sub-analyses for sex-related differences and observed an influence on *UQCC3* and *SDHB* gene expression. *UQCC3* encodes for a ubiquinol-cytochrome C reductase complex assembly protein required for efficient complex III assembly and *SDHB* encodes for a subunit of succinate dehydrogenase, mitochondrial complex II. Both sex and age would be expected to influence differential gene expression in skeletal muscle. Fundamental differences in gene expression profiles between the sexes are known to occur in mammalian species [38]. Aging individuals are known to have elevated skeletal muscle gene expression in pathways regulating cell growth and the immune response, whilst gene expression associated with energy metabolism and mitochondrial function is known to decrease [39,40,41,42].

With regards to MHS sub-phenotypes, baseline comparisons between MHS_hc_ and MHN samples showed far fewer differentially expressed genes to the MHS_h_ vs. MHN comparison. This result was surprising as MHS_hc_ samples have historically generated the strongest IVCT contracture responses and one might expect a greater degree of dysregulation in the resting state. We speculate that the wider age range within our MHS_h_ cohort has contributed to the large number of differentially expressed genes identified. The MHN vs. MHS_hc_ comparison had fewer genes, but we noticed a larger effect size overall, which may reflect differences in IVCT phenotype severity [43]. Only 18 genes were featured in the commonality plot between MHS sub-phenotypes, and they were mostly upregulated (14/18 genes). This was also surprising given that 75% of genes in the “MHN vs. MHS” comparison were downregulated. The two MHS sub-phenotypes appear to have largely unique gene expression profiles at baseline, which may drive the high variability observed in IVCT responses.

Our pre vs. post-IVCT comparisons identified gene expression changes, but the data were somewhat counterintuitive. The IVCT increased gene expression in pro-inflammatory pathways across all phenotypes, and no distinct profile was observed in MHS samples. The IVCT also downregulated OXPHOS gene expression in MHN samples alone, possibly because MHS samples already have depressed OXPHOS gene expression profiles at baseline. There is a question as to whether the halothane and caffeine exposure time/dosage applied during the IVCT was sufficient to trigger relevant change in gene expression in our study. The use of human-cultured myotubes, rather than excised muscle, may be used to refine the experimental design.

Detailed information on physical training and medical history was not available for our analyses and maybe another factor accounting for some of the variability seen between datasets. Increased muscle mass is typically correlated with increased mitochondrial biogenesis—an adaptation from physical training, which can also alter the muscle-fibre-type distribution in skeletal muscle. The distribution of muscle fibre type in the quadriceps of untrained individuals is estimated to be in the range of ~40% type I, and ~60% type II [44]. The distribution of fibre types within each muscle biopsy may influence gene expression since type I and type II have different modes of energy metabolism [45]. In a study of skeletal muscle transcriptomics in rats, high aerobic capacity muscle indicated enhanced tissue oxygenation and vascularization, unlike low-capacity muscle transcriptome, which indicated immune response and metabolic dysfunction relating to inflammation [46]. Similarly, the porcine muscle transcriptomic study by Liu et al. highlighted the importance of mitochondrial oxidative capacity for breed-dependent molecular pathways in muscle cell fibres [25]. On that basis, MHS individuals with a high composition of type I slow-oxidative fibres may be at greater risk of developing myopathic traits with a severe MH phenotype.

Aside from OXPHOS, genes associated with fatty acid metabolism were also dysregulated in MHS samples. Pathway enrichment analysis of MHS baseline samples revealed a downregulation of genes involved in fatty acid metabolism, but our sub-analysis (overlap between MHS_h_ and MHS_hc_) also identified several related genes (*ADIPOQ*, *PLA2G2A*, *MAOA*) upregulated with respect to MHN muscle. This suggests that the dysregulation in MHS fatty acid metabolism is not unidirectional, and the mechanism has additional complexity. Historical data first associated MH susceptibility with defects in fatty acid metabolism after free fatty acid (FFA) accumulation was detected in the mitochondria of MHS pigs [47,48,49]. There was also a suggestion that elevated levels of FFA modulated halothane sensitivity in MHS muscle [50,51,52]. More recently, studies using MH mouse models have reported evidence of lowered fatty acid oxidation, which would likely result in FFA accumulation due to inadequate breakdown [13,14]. This was supported in a metabolomic profiling study that reported elevation in a range of lipid metabolites in human MHS muscle [53]. These data in combination with our transcriptome observations in human MHS muscle strengthen the speculated involvement of fatty acid metabolism in MH.

*RYR1* variants are known to have a pathogenic role in MH, but genotype–phenotype discordance is seen, and genetic susceptibility likely conforms to an oligogenic/threshold model. Interrogation of the burgeoning amount of exome and whole-genome sequence for this disorder, prompted by the transcriptional differences seen in this study, would help in the identification of additional loci contributing to MH genetic susceptibility, and further the development of a non-invasive genetic screen. Investigating nuclear-coded genes that encode proteins involved in respiratory chain complexes would be worthwhile [54]. Age, sex, fitness, and fibre type will all influence transcriptomic profiles in addition to the underlying genetic defects. Nevertheless, the integration of genome sequencing, transcriptomics, and bioinformatics data should prove fruitful to further our understanding of complex phenotypes [55], and predicting potential biomarkers in relation to RYR1-associated disease [56]. Whilst protein studies are needed to follow up observed transcriptional differences, and their full implications with respect to pathological processes remain to be explored, this study has confirmed the importance of mitochondrial metabolism in MH.

## 4. Materials and Methods

### 4.1. Human Muscle Samples

Skeletal muscle biopsies were collected for the IVCT at the Leeds MH Unit, the UK testing centre. Patients gave written informed consent to the study approved by Leeds (East) Research Ethics Committee (reference 10/H1306/70). The IVCTs were conducted according to the protocol outlined by the EMHG [3]. All biopsies were frozen in RNAlater for long-term storage at −80 °C. Depending on the IVCT responses, MHS samples were classified into sub-phenotypes MHS_h_ (responds to IVCT-halothane exposure only) or MHS_hc_ (responds to IVCT-halothane and IVCT-caffeine exposure) for sub-analyses.

### 4.2. RNA Sample Preparation

Total RNA was extracted from vastus medialis muscle biopsies using an IKA T-10 basic ULTRA-TURRAX^®^ homogeniser and a chloroform/isopropanol extraction method. RNA samples used for sequencing were cleaned up using the RNAeasy microkit (QIAGEN, Hilden, Germany) with on-column DNase treatment to remove contaminants. RNA samples used for RT-qPCR were cleaned up using ammonium acetate precipitation. The quality of each RNA sample was assessed using the Agilent 4200 Tapestation (Agilent technologies, Santa Clara, CA, USA), and all samples with integrity scores of RINe >7.0 were classed as good quality and acceptable for further use.

### 4.3. RNA Sequencing Design

We selected 12 unrelated individuals (four MHN, four MHS_h_, and four MHS_hc_) for RNAseq, and total RNA was extracted from their muscle biopsies. The muscle biopsies used for this study had been collected for routine diagnostic IVCT. One segment of the biopsy underwent IVCT, whilst another segment from the same biopsy was gassed continuously with carbogen (95% O_2_, 5% CO_2_) in Krebs–Ringer solution (118.1 mM NaCl, 3.4 mM KCl, 0.8 mM MgSO_4_, 1.2 mM KH_2_PO_4_, 11.1 mM Glucose, 25.0 mM NaHCO_3_, 2.5 mM CaCl_2_, pH 7.4) at room temperature. These pairs of biopsies were then frozen in RNAlater at −80 °C for long-term storage. We chose to use the latter sample type in our study to assess baseline transcriptomic differences.

Where possible, we selected MHS samples based on the following criteria. (1) MHS samples were from patients confirmed through genotyping to carry a familial *RYR1* variant. We chose to use samples with an identified familial *RYR1* variant to help reduce any intra-group variability arising from mechanisms outside of RyR1 function. (2) Each MHS sample should ideally have a strong contracture response to halothane, which we defined as 0.5 g contracture at 2 Vol% halothane). Because of limitations in MHS_h_ sample availability, two MHS_h_ individuals (Patient 6 and 8) had contractures <0.5 g at 2 Vol% halothane, one of which also did not have a confirmed familial *RYR1* variant (Patient 6): these factors were taken into consideration when interpreting results (Table 2). In addition, to reduce age and sex-associated variability, all individuals selected were male and where possible of a similar age.

### 4.4. Library Preparation and Sequencing

The initial quantification of all RNA samples was achieved using the Qubit™ RNA HS Assay Kit. PolyA-enriched mRNA libraries were created using the Truseq Stranded mRNA library preparation kit (Illumina^®^, San Diego, CA, USA), according to manufacturer’s guidelines. Quality and quantity checks of all cDNA libraries were achieved using Agilent D1000 screentape (Agilent technologies, Santa Clara, CA, USA) and the Quant-iT™ PicoGreen^®^ dsDNA assay (Invitrogen, Carlsbad, CA, USA). All cDNA libraries were indexed and pooled together at equimolar concentrations before 150 bp paired-end sequencing was performed on Illumina’s HiSeq^®^ 3000 platform (Illumina, San Diego, CA, USA) by the Leeds NGS facility, achieving an average of 25 M reads per sample.

### 4.5. Differential Gene Expression Analysis

Fastq files were exported and processed using a high-performance computer cluster at the University of Leeds. Technical replicates were combined and trimmed using Cutadapt [57], before FastQC was used to assess data quality [58]. All files were aligned to the human reference genome GRCh38.p13 using the STAR aligner and reads were quantified using featureCounts [59,60]. Read count data were then imported into RStudio and analysed using DESeq2 [61,62]. The *p*-values generated from DESeq2 were adjusted for multiple testing using the Benjamini–Hochberg adjustment [63]. Statistically significant genes were defined by a lenient threshold of adjusted *p*-value < 0.1 to generate larger gene lists and empower subsequent enrichment analyses. The R package ggplot2 was used for data visualisation [64].

### 4.6. Gene Ontology and Pathway Analysis

The online enrichment analysis tool Enrichr was used to functionally annotate lists of differentially expressed genes [65,66]. Pathway analysis terms were taken from the MSigDB_Hallmark_2020 library, and gene ontology data were collected from the GO_Biological_Process_2018, GO_Molecular_Function_2018, and GO_Cellular_Component_2018 libraries. Significantly enriched terms are defined as those with an adjusted *p*-value < 0.05.

### 4.7. RNAseq Validation

Targeted validation of OXPHOS genes from baseline MHN vs. MHS comparisons was investigated on the QuantiStudio™ 7 RT-qPCR system using TaqMan^®^ primers and three housekeeping genes [67]. cDNA was generated using the high-capacity cDNA reverse transcription kit (Thermofisher Scientific, Paisley, UK) from a larger sample cohort: 36 MHN and 36 MHS (18 MHS_h_ and 18 MHS_hc_), using biopsy material from both males and females. Information on subject age, sex, IVCT contracture response, and RYR1/CACNA1S variants are available in Appendix A. Each 20X TaqMan^®^ gene expression assay (Appendix A) was performed in triplicate with a total reaction volume of 10 μL in each well. A standard thermocycling program was used with the following temperature steps and duration: 95 °C for 10 min; 40 cycles of 95 °C for 15 s; 60 °C for 1 min.

### 4.8. Statistics

The mean Ct values for each assay were calculated using the ΔΔCt method and expressed as log2FC [68]. Our datasets were non-parametric, assessed using the Shapiro–Wilk test for normality, and initial statistical analysis was achieved using a combination of the Mann–Whitney and Kruskal–Wallis tests. These data were then subdivided into sex and age groupings to investigate the influence of age and sex on gene expression using the same statistical tests.

## Figures and Tables

**Figure 1 ijms-25-03489-f001:**
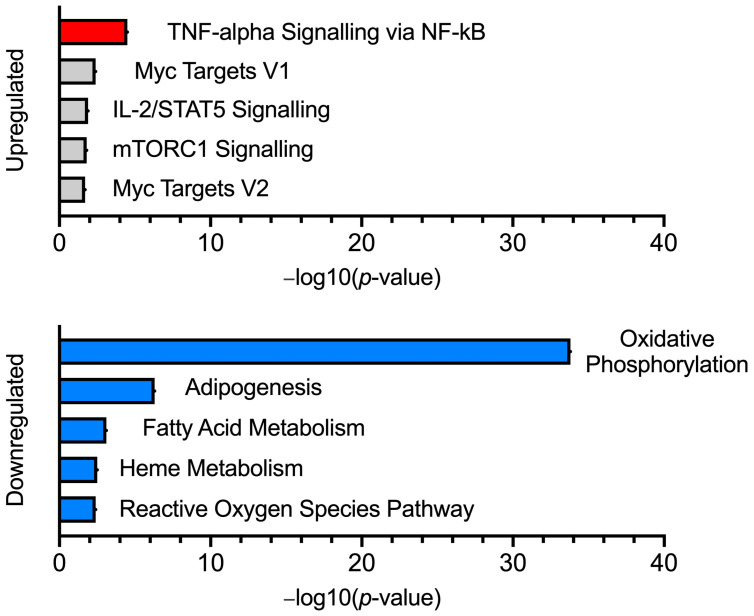
Pathway analysis for MHN vs. MHS. Bar charts featuring the top 5 enriched pathway terms (up and downregulated), generated using genes differentially expressed in the MHS (Pre-IVCT) phenotype. Statistically significant pathway terms are highlighted in red (upregulated) and blue (downregulated).

**Figure 2 ijms-25-03489-f002:**
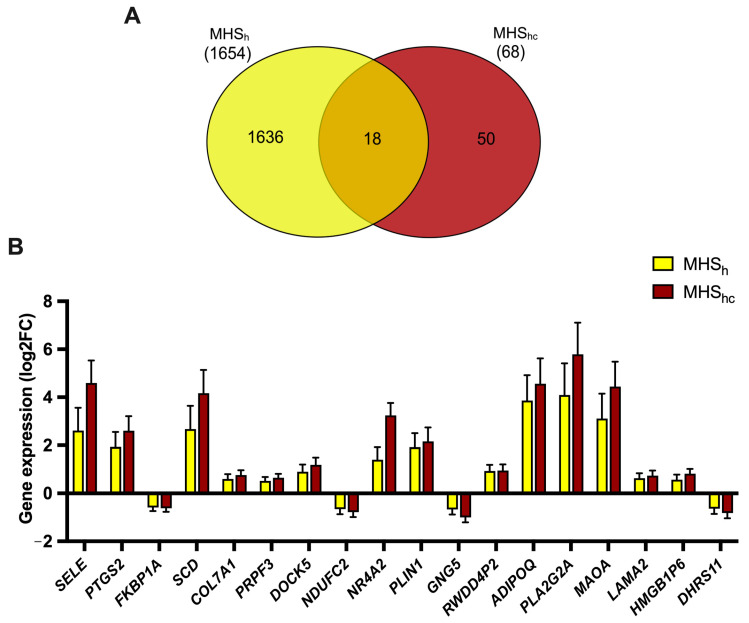
Commonality between MHS_h_ and MHS_hc_ phenotypes: (**A**) Comparisons between ‘MHS_h_ vs. MHN’ and ‘MHS_hc_ vs. MHN’ gene lists were made using a Venn diagram. (**B**) The 18 genes identified in the overlap were plotted against each other to compare their relative expression levels with respect to MHN controls.

**Figure 3 ijms-25-03489-f003:**
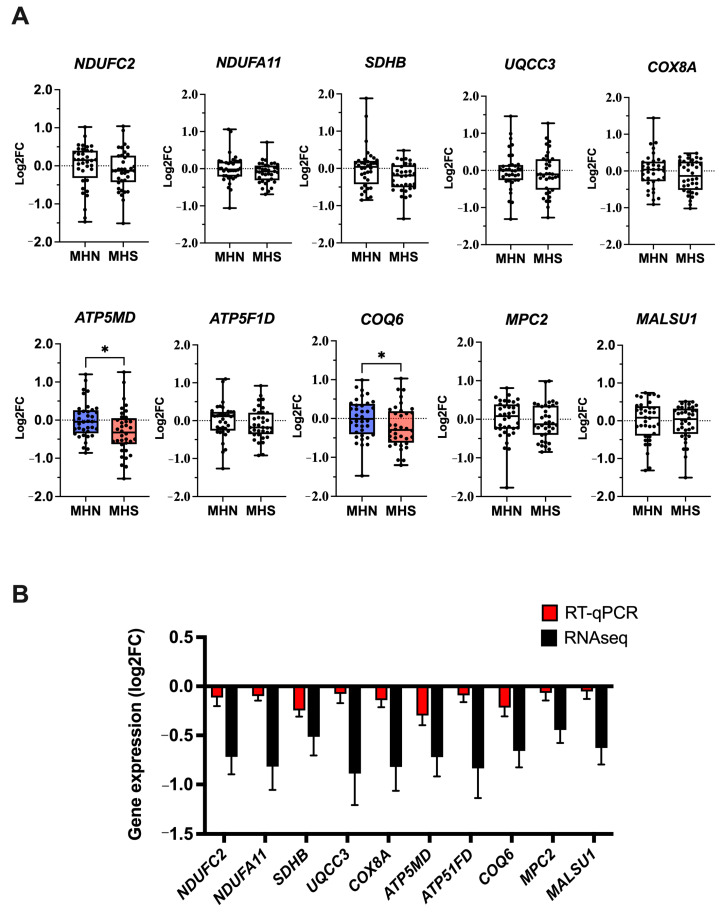
OXPHOS gene expression: (**A**) Boxplots illustrating the expression of 10 OXPHOS genes in MHN (*n* = 36) and MHS (*n* = 36) samples at baseline. Expression data were generated using the QuantiStudio™ 7 RT-qPCR system and presented as log2FC. Statistically significant differences between phenotypes are highlighted using color (MHN:blue, MHS:red) and an * denoting *p*-value < 0.05. (**B**) The log2FC of each gene from RT-qPCR data was compared to the log2FC values obtained through RNAseq.

**Figure 4 ijms-25-03489-f004:**
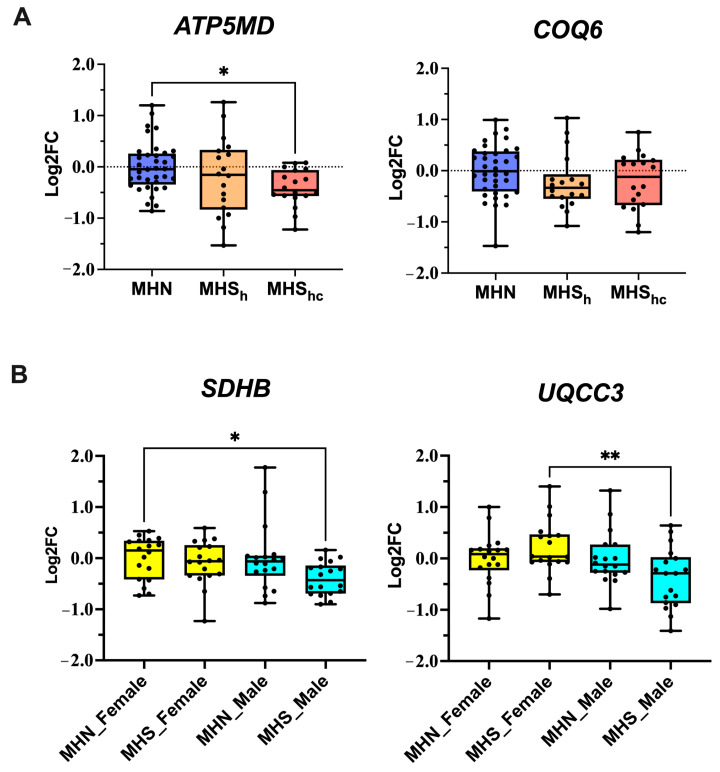
Additional analyses on OXPHOS gene expression: (**A**) Boxplots illustrating differences in *ATP5MD* and *COQ6* expression across MHS sub-phenotypes (MHN: Blue, MHS_h_: Orange, MHS_hc_: Red). (**B**) Boxplots illustrating sex-linked differences in *SDHB* and *UQCC3* gene expression in MHN and MHS, males (light blue) and females (yellow). Statistically significant differences are highlighted with an * denoting *p*-value < 0.05 and ** denoting *p*-value < 0.01.

**Table 1 ijms-25-03489-t001:** Gene ontology results generated by Enrichr. This table shows the top five enriched ontology terms (Biological Processes, Molecular Function, and Cellular Component) generated using the downregulated genes (619 genes) found in MHS muscle. Ontology terms are displayed in order of adjusted *p*-value (significance defined by *p*-value < 0.05).

MHN vs. MHS (Downregulated Genes) Gene Ontology Results				
Biological Processes	*p*-Value	Adjusted *p*-Value	Odds Ratio	Combined Score
Aerobic electron transport chain (GO:0019646)	3.03 × 10^−22^	5.64 × 10^−19^	20.33	1007.15
Mitochondrial ATP synthesis coupled electron transport (GO:0042775)	4.66 × 10^−22^	5.64 × 10^−19^	19.87	976.20
Mitochondrial respiratory chain complex I assembly (GO:0032981)	3.23 × 10^−14^	1.96 × 10^−11^	15.27	474.21
NADH dehydrogenase complex assembly (GO:0010257)	3.23 × 10^−14^	1.96 × 10^−11^	15.27	474.21
Mitochondrial respiratory chain complex assembly (GO:0033108)	1.27 × 10^−13^	6.17 × 10^−11^	10.36	307.73
**Molecular function**	** *p* ** **-value**	**Adjusted *p*-value**	**Odds Ratio**	**Combined score**
Oxidoreduction-driven active transmembrane transporter activity (GO:0015453)	1.35 × 10^−20^	6.05 × 10^−18^	22.50	1029.19
Proton transmembrane transporter activity (GO:0015078)	6.38 × 10^−11^	1.43 × 10^−8^	16.20	380.39
NADH dehydrogenase (quinone) activity (GO:0050136)	1.71 × 10^−10^	1.91 × 10^−8^	17.53	394.25
NADH dehydrogenase (ubiquinone) activity (GO:0008137)	1.71 × 10^−10^	1.91 × 10^−8^	17.53	394.25
Active ion transmembrane transporter activity (GO:0022853)	1.88 × 10^−9^	1.68 × 10^−7^	13.44	269.96
**Cellular component**	** *p* ** **-value**	**Adjusted *p*-value**	**Odds Ratio**	**Combined score**
Mitochondrial inner membrane (GO:0005743)	4.55 × 10^−42^	1.06 × 10^−39^	10.42	991.95
Mitochondrial membrane (GO:0031966)	8.20 × 10^−42^	1.06 × 10^−39^	8.22	777.65
Organelle inner membrane (GO:0019866)	1.97 × 10^−38^	1.70 × 10^−36^	9.35	811.84
Mitochondrial matrix (GO:0005759)	2.17 × 10^−18^	1.40 × 10^−16^	5.51	223.92
Mitochondrial respiratory chain complex I (GO:0005747)	4.31 × 10^−13^	1.86 × 10^−11^	18.76	534.17

**Table 2 ijms-25-03489-t002:** Summary of patients’ details selected for RNAseq. This table summarises the gender, biopsy age, MH status, IVCT contracture strengths (halothane and caffeine), and familial RYR1 variant for everyone selected for RNAseq.

Patient	Gender	Age at Time of Biopsy	MH Status	Contracture at 2% Halothane	Contracture at 2 mM Caffeine	*RYR1* Family Variant
1	M	13	MHN	0	0	Negative
2	M	11	MHN	0	0	Negative
3	M	19	MHN	0	0	Negative
4	M	19	MHN	0	0	Negative
5	M	12	MHS_h_	1.15 g	0	c.7007G>A
6	M	39	MHS_h_	0.25 g	0	Unknown
7	M	30	MHS_h_	0.75 g	0	c.4293G>A, c.7879G>A
8	M	12	MHS_h_	0.25 g	0	c.5183C>T
9	M	18	MHS_hc_	0.65 g	2.1 g	c.12700G>C
10	M	11	MHS_hc_	1.8 g	0.6 g	c.7292A>T
11	M	16	MHS_hc_	1.6 g	0.4 g	c.14201G>A
12	M	14	MHS_hc_	0.9 g	0.6 g	c.6617C>T

## Data Availability

All source sequencing data for this study are openly available at time of publication at NCBI Sequence read archive (SRA) accession: PRJNA1075675.

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
