# Peer review of "An Association between OXPHOS-Related Gene Expression and Malignant Hyperthermia Susceptibility in Human Skeletal Muscle Biopsies"

_ijms, 2024, doi:10.3390/ijms25063489_

Round 1

Reviewer 1 Report

Comments and Suggestions for Authors

This manuscript by Chang et al. extends previous analysis on the transcriptomic changes in malign hyperthermia syndrome (MHS) using patient muscle biopsy samples and trying to distinguish between subtypes of the phenotype. It reaches the conclusion that changes in mitochondrial metabolism can be important in this disorder.

- The results, although interesting, are somehow limited by the use of a single parameter analyzed (transcript levels) and by the fact that there seems to be some incongruencies and lack of correlations. The more important, in my opinion, are two:
1) lack of congruency between the bulk RNAseq data and those of the RT-qPCR method used to “validate” the former. Thus, only 2 out of the 10 genes chosen for that validation where statistically confirmed to be downregulated when the sample size was increased from 4 controls (MHN) and 8 MHS to 36 in each group. And in one of these two genes (ATP5MD) the significance in the difference seems to be driven by only one of the subphenotypes (Figure 4A). The small difference in mean IVCT contracture response (0,914 vs 0,747g) can justify the discrepancy in these results? Since the RT-qPCR seems a more direct and reliable technique and the sample number is much higher, is the conclusion of the study compromised? Why the authors did not analyze by RT-qPCR more genes, for example complex I (CI) genes?

2) lack of correlation between the expression changes in OXPHOS genes reported and the OXPHOS activities found in previous works. While 18 CI subunits are found downregulated and CI deficiency is suggested (lines 219-220 of this manuscript) no clear CI activity reduction (per muscle mass or as FCR) has been found in previous articles from the group using either patient muscle biopsies (Chang, 2019) or muscle from an MHS mouse model (Chang, 2020).

As the authors suggest, further proteomic and functional studies are needed to give strength to the correlations found in the transcriptomic analysis.

Other points:

- The number of samples used in experiment 2.2 is not clear. I assume that it is 4 for each group, but it should be more clearly indicated. The same applies to section 2.4.

- Part of the results in section 2.2 are surprising and not explained: a) very few genes changing in the MHShc cohort compared with MHSh, b) the fact that 75 %are downregulated genes but the commonality plot between both, out of 18 genes with overlapping levels only 4 (less than 25%) are downregulated and only 1 is an OXPHOS subunit.

- Are the sex differences (only reported for two genes, small in terms of log2FC and between two distinct pairs of samples), relevant?

- The ICVT induced changes (section 2.4.) are also surprising and their significance is relative as admitted by the authors (lines 305-307).

- The figure S1 mentioned in section 2.4 is not found among the supplemental information provided.

Error in line 175: “... on IVCT ...” should be “... of IVCT …”

Reviewer 2 Report

Comments and Suggestions for Authors

The manuscript concerns a study of expression patterns in muscle biopsies from malignant hyperthermia susceptible patients. The presented data is of interest, yet the manuscript is not acceptable for publication in its current form.

The abstract needs substantial revision, as it is not informative enough as it stands. No mention is made of the division h and hc which are referred to only as ‘sub-phenotypes’, patient numbers are not given, no true results are included, no mention is made of validation experiments. After some serious thinking of what the authors aimed, found and conclude, this abstract needs to be rewritten from scratch.

The different subgroups of patients need to be defined (including what h c responding means) in the M&M section. What muscle was sampled? This is important for the interpretation of oxphos results, as different muscle groups have different mitochondrial load. The meaning of the RYR1 variant column in table 2 is confusing. List only causal gene mutations or unknown. Supplementary files are not referred to properly in the text.

Discussion does not touch upon the complex I oriented downregulation. Any explanation speculation here? Validation of findings is limited, done only on the mRNA level and not protein level. Only 2 in 10 checked factors were downregulated with statistical significance, and only one of these concerned a complex I-associated factor. The authors cite their former study showing complex II dysfunction, is this a discrepancy? Why were the findings not confronted with enzymatic analysis of oxphos complex activities as a diagnostic test often performed in human muscle biopsies?

The authors put forward ROS as a mechanism compromising mitochondrial function, however, I do not see ROS detoxification pathways activated in the RNAseq. Also, the discussion talks about the genotype-phenotype discordance in patients but does not really focus on that aspect. The patients have not been fully described at the clinical level, and are genetically diverse or not characterized. Overall, the number of patients analyzed is fairly limited. Such a smaller cohort would have allowed to do a more thorough characterization, which would have been of more value to the scientific community.  Also, when subdividing based upon therapeutic outcome h c, the choice seems to be made to focus on the therapeutic aspect. This point is not developed enough in the discussion.  

The text needs a general concluding remarks. What have we learned from this study for medical science and clinical practice?

Comments on the Quality of English Language

English language is fine

Round 2

Reviewer 1 Report

Comments and Suggestions for Authors

Although many of the justifications in the answer to the main concerns are debatable and I am not fully convinced, I think the article can be published and add some useful information to the field.

Reviewer 2 Report

Comments and Suggestions for Authors

Revisions have been carefully made, hence the manuscript has much improved.

Comments on the Quality of English Language

English language is fine